# Characterization of *Staphylococcus aureus* Isolates from Bovine Mastitis and Bulk Tank Milk: First Isolation of Methicillin-Susceptible *Staphylococcus aureus* in Japan

**DOI:** 10.3390/microorganisms10112117

**Published:** 2022-10-26

**Authors:** Ryota Miyazawa, So Shimoda, Keiichi Matsuda, Ryuta Tobe, Tasuke Ando, Hiroshi Yoneyama

**Affiliations:** 1Laboratory of Animal Microbiology, Department of Animal Science, Graduate School of Agricultural Science, Tohoku University, 468-1, Aramaki Aza Aoba, Aoba-ku, Sendai 980-8572, Miyagi, Japan; 2Livestock Medicine Training Center, Miyagi Prefecture Agricultural Mutual Aid Association, 39-4, Oohira Hirabayashi, Oohira-Village, Kurokawagun 981-3602, Miyagi, Japan

**Keywords:** *Staphylococcus aureus*, mastitis, ST398, methicillin-sensitive *Staphylococcus aureus* (MSSA), *spa* typing

## Abstract

*Staphylococcus aureus* is one of the most important pathogens in humans as well as in livestock. Particularly, bovine mastitis caused by *S. aureus* is a serious issue in dairy farms due to disease recurrence. Here, cases of *S. aureus*-mediated intramammary infection occurring in the Miyagi Prefecture in Japan were monitored from May 2015 to August 2019; a total of 59 strains (49 from bovine milk and 10 from bulk milk) were obtained from 15 dairy farms and analyzed via sequence-based typing methods and antibiotic susceptibility tests. Two pairs of isolates were determined as recurrence cases from the same cows in distinct farms. The sequence type (ST), *spa* type, and *coa* type of each pair were the same: one pair showed ST705, t529, and VIb and the other showed ST352, t267, and VIc. In addition, the possession of toxin genes analyzed of each pair was exactly the same. Furthermore, seven oxacillin-sensitive clonal complex 398 isolates were obtained from a single farm. This is the first confirmed case of a Methicillin-Sensitive SA (MSSA) ST398 strain isolated from mastitis-containing cows in Japan. Our findings suggest that nationwide surveillance of the distribution of ST398 strains in dairy farms is important for managing human and animal health.

## 1. Introduction

Mastitis, an inflammatory response in the mammary gland primarily caused by bacteria, represents one of the most important diseases in the dairy industry [1], as it causes annual economic losses of over $2 billion in the United States [2] and £300 million in the United Kingdom alone [1]. In addition, mastitis affects not only animal welfare [1,3] but also human health as a result of increased antimicrobial use [4].

Mastitis control procedures such as appropriate milking procedures, teat disinfection after milking, segregation of infected animals, and prevention of the introduction of infected animals on farms have been developed. Nevertheless, antibiotic treatment remains an important and effective method for treating infections [1,5,6].

Among the mastitis-causing pathogens, *Staphylococcus aureus*, which infects the mammary gland, is one of the most important ones. The physiological characteristics of *S. aureus* make antibiotic treatment of bovine mastitis difficult because the bacterium survives after internalization in phagocytic cells [7,8] and forms intramammary biofilms [5], which may require repeated antibiotic therapy. The use of large amounts of antibiotics in the livestock industry can lead to the emergence of antibiotic-resistant bacteria, among which methicillin-resistant *S. aureus* (MRSA) represents a major public health concern, as it shows resistance to many clinically important antibiotics [9,10]. Although the emergence of MRSA in a bovine mastitis case is infrequent [11], isolates with MRSA characteristics have been reported [12,13]. Accordingly, the information on the antibiotic susceptibility of *S. aureus* strains isolated from bovine mastitis cases is important.

*S. aureus* is a major human pathogen that can also cause an array of infections in various economically important livestock [9]. Owing to this pathological implication, a number of *S. aureus* isolates from various animals, including humans, have been biochemically and genetically characterized to determine their phylogenetic relationship; *S. aureus* strains have co-evolved with a single host species and adapted to colonize a specific host animal [7,9,14]. Recent genome sequencing-based approaches have further strengthened the phylogenetic relationships of various *S. a**ureus* strains isolated from different animals. Specifically, multi-locus sequence typing (MLST) analysis has shown that a specific sequence type (ST) and clonal complex (CC), which is a group of closely related STs, are associated with distinct animal hosts [5,9]. For example, *S. aureus* strains isolated from intramammary infections of ruminants mostly belong to CC97, CC133, CC130, CC126, and CC705 [5,9]. Additionally, most strains isolated from poultry and rabbit infections belong to ST5 and ST121, respectively [5,9].

*S. aureus* isolates produce various virulence factors, toxins, bacterial components that adhere to its host cells/tissues, and immune evasion molecules, which play an important role in their pathogenicity [10,15]. Bacterial surface factors are involved in the initial stages of the infection. The latex agglutination test is a convenient and widely used diagnostic test to confirm putative *S. aureus* isolates and to assess the presence of *S. aureus* virulence factors by detecting characteristics of their surface-anchored proteins [16].

The presence of MRSA strains in livestock represents a public health concern. In particular, the first isolation of MRSA strains from pigs, which belong to the ST398 genotype [17], has renewed the interest of researchers in livestock-associated *S. aureus*. Subsequently, MRSA strains belonging to the ST398 lineage have been isolated from calves and poultry [18,19]. As ST398 MRSA strains are known to have a wide host specificity, including humans [9], the surveillance of *S. aureus* strains isolated from livestock animals is important from a public health perspective; this is particularly true for Japan, as ST398 strains have not been isolated from cows so far [12]. In this study, we described the distribution of *S. aureus* strains isolated from cows between 2015 to 2019 using sequence-based typing methods, analyzed the isolates for their presence of various virulence factors by PCR and latex agglutination tests, and determined the susceptibility of the isolates to antibiotics. This study is the first to isolate methicillin-sensitive ST398 strains in Japan from a single farm out of 15 dairy farms surveyed in the central region of Miyagi Prefecture.

## 2. Materials and Methods

### 2.1. Bacterial Strains and Growth Conditions

A total of 59 *S. aureus* strains were isolated from the milk of cows (Holstein) with mastitis (49 strains) and bulk milk (10 strains) that was collected from a combined milk of respective farm using an aseptic syringe by a licensed practicing veterinarian from Miyagi Prefecture, Japan, between June 2015 and August 2019. Milk samples (10 μL of individual mastitis cow milk and 50 μL of bulk milk) were spread on trypticase soy agar (1.5%, *w*/*v*) (TSA; Nissui Pharmaceutical Co., Tokyo, Japan) with 5% sheep blood (Becton Dickinson Co., New Jersey, NY, USA) and incubated at 37 °C for 24 h. Peracute mastitis was diagnosed from integrated diagnostic symptoms of an individual cow that showed local symptoms (swelling and induration of udders) and systemic symptoms (high temperature, loss of forage intake, inadequate circulation, and difficulty in standing). Veterinarian diagnosis of all the collected milk was performed via the modified California mastitis test using a PL tester (Nippon Zenyaku Kogyo Co., Tokyo, Japan) and temperature measurement of individual cows. Isolates were grown in tryptic soy broth (TSB; Nissui Pharmaceutical Co., Tokyo, Japan) at 37 °C with shaking or on TSA at 37 °C aerobically. Procedures for handling cows conformed to the Basic Guidelines for the Research Use of Industrial Animals in Miyagi Prefecture Agricultural Mutual Aid Association. *S. aureus* ATCC29213 was used as a control strain in the MIC determination.

### 2.2. DNA Isolation

DNA was isolated from cells grown in TSB containing 3% (*w*/*v*) glycine by lysing the cells with acromopeptidase (Wako Pure Chemical Co., Osaka, Japan) and lysozyme (Wako Pure Chemical Co.) as described previously [20]. Briefly, cells (2 mL) at the mid-logarithmic growth phase were collected by centrifugation (16,100× *g*, 2 min, 23 °C) and suspended in a solution (300 μL) containing 0.3 M sucrose, 25 mM Tris-HCl, and 25 mM EDTA (pH 8.0). After centrifugation as above, cells were resuspended in 300 μL of the same solution. Subsequently, 50 μL of 2.4 mg/mL acromopeptidase dissolved in the same solution and 8 mg/mL lysozyme dissolved in the same solution were added to the cells to make final concentrations of 0.3 mg/mL and 1 mg/mL, respectively, and incubated at 37 °C overnight. The cells were then mixed with 100 μL of sodium dodecyl sulfate (final concentration, 2%) and incubated at 65 °C for 10 min, followed by extraction with a mixture of phenol and chloroform (1:1). An aqueous phase was then obtained by centrifugation (16,100× *g*, 5 min, 23 °C). After precipitation of DNA with 3 M sodium acetate (50 μL) and isopropanol (330 μL), the precipitate was washed with 70% ethanol, and the final air-dried DNA sample was dissolved in a solution (20 μL) containing 1 mM Tris-HCl and 0.1 mM EDTA (pH 8.0).

### 2.3. Genotyping of Isolates

An MLST analysis was performed as described previously [21]. Briefly, the gene fragments of *arcC*, *aroE*, *glpF*, *gmk*, *pta*, *tpi*, and *yqiL* were amplified using specific primers (Appendix A), bacterial isolate DNA as the template, and TaKaRa EX Taq polymerase (Takara Bio Co., Shiga, Japan). The amplified products were purified using the FastGene Gel/PCR Extraction Kit (Nippon Genetics Co., Tokyo, Japan), and the nucleotide sequence was determined via the dideoxy chain termination method using a 3500 Genetic Analyzer (Thermo Fisher Scientific, Waltham, WA, USA) [22]. Subsequently, the allele number and ST were determined from the *S. aureus* MLST website (https://pubmlst.org/saureus/ (accessed on 29 August 2022)). The *spa* typing was performed as described previously [23]. Briefly, the nucleotide sequence of *spa* in individual isolates was amplified using specific primers (Appendix A) and analyzed using Ridom SpaServer (https://www.spaserver.ridom.de/ (accessed on 22 August 2022)). Classification of the staphylocoagulase type was performed via multiplex polymerase chain reaction (PCR), following a protocol described previously [24], using specific primers (Appendix A) to detect the coagulase gene (*coa*) type I to VIII and subtypes VIa, VIb, and VIc.

### 2.4. Detection of Toxin and Tetracycline Resistance Genes

The presence of four leucocidins, four hemolysins, 13 enterotoxins, and toxic shock syndrome toxin genes was detected via PCR using specific primers (Appendix A), as described previously [25]. The presence of tetracycline resistance genes *tetG*, *tetK*, *tetL*, *tetM*, *tetO*, and *tetS* was determined via PCR analysis using specific primers (Appendix A) as previously described [26].

### 2.5. Latex Agglutination Test

A latex agglutination test was performed using the PS Latex Kit (Eiken Kagaku Co., Tokyo, Japan) according to the manufacturer’s instructions. Briefly, individual isolates grown on TSA were suspended in approximately 50 μL of 0.85% NaCl on a glass slide, and then a drop of latex solution (approximately 50 μL) was added and gently mixed. Agglutination scores were determined as +++, ++, and + for agglutination observed within 20 s, 20–40 s, and 40–60 s, respectively.

### 2.6. Minimum Inhibitory Concentration (MIC) Determination

The MICs for ampicillin, chloramphenicol, cefazolin, erythromycin, kanamycin, oxacillin, streptomycin, and tetracycline were determined via the agar dilution method following the standard protocols of the Clinical and Laboratory Standards Institute [27]. Briefly, each strain was grown at 37 °C overnight in Mueller Hinton broth (Beckton, Dickinson and Company, Sparks, MD, USA) with shaking. Cells were then inoculated on Mueller Hinton agar medium (Beckton Dickinson, Sparks, USA) without or with 2% NaCl (for oxacillin) containing 2-fold serially diluted antibiotics by using a microplanter (Model MIT-P, Sakuma Co., Tokyo, Japan) (appropriately 10^4^ CFU/spot). The MIC was defined as the lowest concentration of antibiotics that inhibited the growth of the test strain after overnight incubation at 37 °C. We repeated the MIC determination at least twice for each antibiotic and obtained reproducible results.

## 3. Results

### 3.1. Genetic Typing of S. aureus Isolates

#### 3.1.1. Sequence Type and Clonal Complex

A total of 59 *S. aureus* strains isolated from intramammary infection cases (49 strains) and bulk milk (10 strains) from 15 dairy farms were analyzed using the MLST (ST and CC), *spa* typing, and *coa* typing methods. The typing results (ST and CC), together with veterinarian diagnosis, are presented in Table 1.

A total of seven STs could be typed, among which eight isolates were found to not belong to any existing ST (as of September 2022); therefore, new STs denoted as NT1 and NT2 were described in this study. All strains were grouped into three CCs: CC97 (*n* = 28), CC705 (*n* = 24), and CC398 (*n* = 7). Notably, seven strains belonging to CC398 were isolated: they were repeatedly isolated from June 2015 to May 2019 on a specific farm (farm B). Among them, one strain belonged to ST4898, which has recently been isolated in the Czech Republic (Jolley et al. *Wellcome Open Res.* 2018, 3:124 [version 1; referees: 2 approved]) [28].

#### 3.1.2. *spa* Type

The *spa* typing analysis revealed six different *spa* types: t267 (19 strains), t359 (5 strains), t529 (22 strains), t865 (1 strain), t1201 (5 strains), and t3934 (7 strains) (Table 1). When a combination of CC and *spa* typing was evaluated, the majority of the CC705 strains (21 out of 24 strains) belonged to t529 (Table 2). However, results of the *spa* typing of CC97 strains tended to vary. All CC398 strains showed the same *spa* type, t3934 (Table 2).

#### 3.1.3. *coa* Type

The staphylocoagulase type of the 59 isolates was classified based on differences in the nucleotide sequence of *coa* via a multiplex PCR method [20]. All isolates were assigned to three *coa* types: VIb (22 strains), VIc (30 strains), and VII (seven strains) (Table 1).

### 3.2. Characterization of S. aureus Isolates

#### 3.2.1. Leucocidins, Hemolysins, and Superantigenic Toxins

We examined whether the isolates harbored genes coding for leucocidins, hemolysins, and superantigenic toxins. In total, eight genes (*lukS*, *lukF*, *sea*, *seb*, *sed*, *see*, *seh*, and *sej*) out of the 22 genes evaluated were not detected; however, four genes (*hla*, *hlb*, *hld*, and *hlg*) were detected (Table 3 and Appendix A). The combination of the genes present was classified into five groups (Table 3). Among the toxin genes detected, *lukD* and *lukM* were found in 89.8% of isolates in this study (Table 3 and Appendix A).

In relation to the *coa* type results, almost all *coa* VIb subtype strains (19 out of 22 VIb strains) harbored superantigenic genes (*sec*, *seg*, *sei*, *sel2*, *sem*, *sen*, *seo*, and *tst*). In contrast, 20 out of 30 VIc strains did not carry all these toxin genes, except for the SA9 isolate, which harbored all of the superantigenic genes tested. Notably, nine *coa* VIc subtype strains possessed only the *seo* gene, but not other genes (*sec*, *seg*, *sei*, *sel2*, *sem*, *sen*, and *tst*) (Appendix A).

#### 3.2.2. Latex Agglutination Properties

To test whether *S. aureus* isolates possess cell surface-anchored proteins, we assessed the latex agglutination phenotype of the isolates. Twenty-two isolates (37%) were latex test-negative. Notably, all latex-negative strains appeared to belong to *coa* type VIb (Table 4). In contrast, all isolates belonging to *coa* type VII and subtype VIc were latex test-positive to a certain extent. Furthermore, although the number of CC398 (*coa* type VII) isolates was small, the latex test reaction in strains of this type tended to be stronger (one + +, six + + +) than that observed in strains belonging to *coa* subtype VIc, where the proportion of isolates with score + + + in *coa* subtype VII and VIc was 85.7% (six out of seven isolates) and 40% (six out of 30 isolates), respectively.

#### 3.2.3. Antibiotic Susceptibility

To test antimicrobial susceptibility phenotypes of the *S. aureus* isolates, we determined their antibiotic susceptibility to ampicillin, oxacillin, cefazolin, chloramphenicol, tetracycline, erythromycin, kanamycin, and streptomycin. All isolates were susceptible to oxacillin (MIC, 0.0625–0.5 μg/mL), cefazolin (0.125–0.5 μg/mL), erythromycin (0.25–0.5 μg/mL), kanamycin (2–4 μg/mL), and streptomycin (4–16 μg/mL) (Appendix A). Notably, all CC398 strains tested showed resistance to tetracycline, MICs of which, 32 or 64 μg/mL, were larger than the breakpoint of this antibiotic [29]. In contrast, all CC97 and CC705 strains were susceptible to tetracycline (Appendix A). All CC398 strains isolated in this study possessed *tetM* (Appendix A). All CC398 strains, except for SA21, and one CC705 strain, SA40, showed resistance to ampicillin (8–16 μg/mL). The other strains, including one CC398 strain, SA21, showed an MIC ≤ 0.5 μg/mL for ampicillin (Appendix A).

## 4. Discussion

The presence of MRSA strains in livestock is a public health concern. Therefore, we performed the molecular characterization of *S. aureus* isolated from bovine mastitis cases in the Miyagi Prefecture, Japan. Genetic typing showed that the predominant isolates analyzed were ruminant-associated: 28 and 24 isolates were grouped into clonal lineages CC97 and CC705, respectively, which is consistent with the results of earlier studies wherein *S. aureus* strains in lineages CC97 and CC705 have been reported as the major causative agents of bovine mastitis in Japan and in other countries [30,31,32,33]. The results of the *spa* typing of CC97 strains tended to vary, which is consistent with the results reported in previous studies [31,34]. Our findings were in agreement with those of previous studies showing that the majority of *S. aureus* isolates from bovine raw milk are classified into either *coa* subtype VIb or VIc [24]. Additionally, the pattern of possession of *lukD* and *lukM* was found to be similar to that reported in earlier studies wherein *lukS* and *lukF* genes encoding Panton–Valentine leucocidin were absent, and *lukD* and *lukM* were frequently identified in bovine isolates [31,35]. In addition, almost all *coa* subtype VIc *S. aureus* isolates, except for isolate SA9, belong to toxin group 2 or 3 (Appendix A), in which only a superantigenic toxin *seo* was found in the group 3, which is consistent with the previous study [24].

Notably, we found that seven strains belonging to CC398 (six ST398 clones and one ST4894 clone) were isolated from both cow milk (five isolates) and bulk milk (two isolates) from the same farm (farm B). Among the five ST398 isolates from cow milk, three strains (SA21, SA22, and SA23) were isolated from distinct quarters of the same cow on the same day, indicating that this *S. aureus* type was found in three out of 49 cows afflicted with mastitis (6.8% of cows tested in this study). Recent in-depth molecular typing experiments have revealed that most *S. aureus* strains are host-specific; CC97 and CC705 strains are commonly associated with cows [9,33]. In contrast, ST398 clones have a broad host specificity [9,11]. A serious concern regarding public health is the frequent association of ST398 strains with the methicillin resistance phenotype; MRSA isolates (MRSA ST398) are widely disseminated in pigs worldwide and can colonize multiple host species, including cows, sheep, poultry, horses, and humans [9,36]. However, *S. aureus* ST398, which causes bovine mastitis, has not yet been detected in Japan [12]. Notably, five ST398 isolates obtained from three mastitis cows were oxacillin-susceptible (classified as MSSA) (Appendix A). Furthermore, all these ST398 isolates showed tetracycline resistance, which is consistent with the results of other studies showing that MRSA ST398 isolates from bovine mastitis [13,25] and CC398 strains isolated from humans [37] generally possess a tetracycline resistance phenotype. Accordingly, it is worth noting that five ST398 isolates from cows found in this study were susceptible to oxacillin, as MSSA ST398 clones are commonly found in pigs, but not in other animals, including cows [10,38]. The tetracycline resistance phenotype of the CC398 isolates may be attributed to the horizontal gene transfer of tetracycline resistance genes [10,36]. As MSSA ST398 clones were isolated only from farm B, and strains in the lineage ST398 are rarely found—if present at all—in Japan, it can be speculated that these ST398 strains are not prevalent in farms in the Miyagi area and could have been transmitted from other animals and/or humans. This speculation is supported by the fact that all CC398 strains isolated from farm B do not possess *lukD* and *lukM*, which are strongly associated with bovine strains. To the best of our knowledge, this study is the first to report the isolation of ST398 lineage of *S. aureus* from bovine mastitis cases in Japan.

We found in this study that two pairs of isolates were obtained from the same quarter of the respective cow at different lactation periods: SA16 and SA19 (CC705/ST705 strains) from farm C, and in the same lactation period, SA11 and SA18 (CC97/ST352 strains) from farm E, in which SA19 and SA18 were isolated after the disappearance of the previous clinical mastitis symptoms caused by SA16 and SA11, respectively. Each pair showed the same genetic typing results: SA16 and SA19 were ST705 (CC705), *spa* type t529, and *coa* type VIb; SA11 and SA18 were ST352 (CC97), *spa* type t267, and *coa* type VIc. In addition, the classification based on possession of toxin genes was the same for each pair: SA16/SA19 and SA11/SA18 pairs belonged to groups 5 and 2, respectively (Table 3). Thus, it was speculated that the recurrence of mastitis in the respective cow was caused by the same *S. aureus* clone, irrespective of cefazoline treatment in both cases, a phenomenon of which was documented previously [39].

We also found four isolates that showed peracute symptoms, one from farm D (SA38) and three from farm G (SA31, SA33, and SA49). It is interesting to note that SA38 was latex agglutination test-negative in contrast to other isolates (SA31, SA33, and SA49) that showed strong latex agglutination phenotype (Table 4). Although the reason of the latex-negative response of SA38 remains unclear, it is speculated that genetic change and/or variation of the membrane-associated virulence factors, described as microbial surface components recognizing adhesive matrix molecules (MSCRAMM), of this isolate could cause the latex-negative phenotype, as reported previously [40]. Further in-depth analysis is needed to address this issue.

This study covered a limited region of the Miyagi Prefecture (central part of Miyagi) and showed that typical CC97 and CC705 strains, which are known to be associated with bovine mastitis worldwide and in Japan, were isolated from cows with clinical symptoms and from bulk milk. This implies that typical mastitis-causing *S. aureus* groups are distributed in this limited dairy farm area. Furthermore, a key finding of this study was that MSSA CC398 strains (six ST398 clones and one ST4894 clone) were isolated from bovine mastitis and bulk milk from a single farm out of 15 farms in a limited area of Miyagi. Although the isolation of these MSSA ST398 strains from cow milk is rare, this finding suggests the possibility that ST398 lineage clones may be distributed in dairy farms in Japan to a greater extent than that estimated previously.

ST398 clones are known to infect multiple host species, including bovines and human [9]. MSSA *S. aureus* can evolve into MRSA by acquiring *SCCmec* [41]. In addition, CC398 MSSA isolates that were reported to cause bloodstream infections in a French hospital are more virulent than non-CC398 strains [42]. Hence, epidemiological surveillance and continuous monitoring of the prevalence of CC398 *S. aureus* clones in dairy farms in Japan are required.

## 5. Conclusions

In this study, we analyzed a total of 59 *S. aureus* strains isolated from intramammary infection cases (49 strains) and bulk milk (10 strains) between 2015 to 2019 in Miyagi Prefecture in Japan using sequence-based typing methods. Most importantly, CC398 strains (six ST398 clones and one ST4894 clone) were isolated from bovine mastitis and bulk milk from a single farm (out of 15 farms), which appeared to be oxacillin-sensitive. This is the first confirmed case of methicillin-sensitive *S. aureus* (MSSA) ST398 strains isolated from mastitis-containing cows in Japan. Our findings therefore suggest that nationwide surveillance and continuous monitoring of the prevalence of ST398 strains in dairy farms in Japan is important for managing human and animal health.

## Figures and Tables

**Table 1 microorganisms-10-02117-t001:** Genetic typing of *S. aureus* isolates with veterinarian diagnosis.

No.	Isolate	Farm	Date of Isolation (YYYY. MM. DD)	Source of Milk ^a^	Veterinarian Diagnosis		Typing			*coa* Type
					P.L Tester ^b^	Temp. ^c^	ST	CC	*spa*	
1	SA3	A	2015. 6. 9	Bulk	−	ND	705	705	t529	VIb
2	SA14	A	2015. 10. 13	Bulk	−	ND	705	705	t529	VIb
3	SA28	A	2016. 7. 11	Bulk	−	ND	705	705	t529	VIb
4	SA4	B	2015. 6. 9	Bulk	−	ND	4894	398	t3934	VII
5	SA21 ^d^	B	2016. 4. 5	Individual	±	38.6	398	398	t3934	VII
6	SA22 ^d^	B	2016. 4. 5	Individual	±	38.6	398	398	t3934	VII
7	SA23 ^d^	B	2016. 4. 5	Individual	±	38.6	398	398	t3934	VII
8	SA27	B	2016. 7. 11	Bulk	−	ND	398	398	t3934	VII
9	SA39	B	2017. 10. 13	Individual	++	40.9	398	398	t3934	VII
10	SA58	B	2019. 5. 19	Individual	+	40.0	398	398	t3934	VII
11	SA59	B	2019. 5. 19	Individual	+	40.0	705	705	t529	VIb
12	SA5	C	2015. 6. 21	Individual	++	ND	NT1	705	t1201	VIb
13	SA16 ^e^	C	2015. 10. 14	Individual	++	40.8	705	705	t529	VIb
14	SA19 ^e^	C	2016. 3. 12	Individual	++	39.6	705	705	t529	VIb
15	SA20	C	2016. 3. 16	Individual	+	38.3	705	705	t529	VIb
16	SA29	C	2019. 5. 19	Individual	+	39.0	705	705	t529	VIb
17	SA30	C	2016. 11. 14	Individual	++	39.9	705	705	t529	VIb
18	SA35	C	2017. 6. 5	Individual	++	38,5	705	705	t529	VIb
19	SA36	C	2017. 6. 5	Individual	++	38.5	705	705	t529	VIb
20	SA6	D	2015. 7. 6	Individual	++	ND	352	97	t267	VIc
21	SA7	D	2015. 7. 14	Bulk	−	ND	NT1	705	t1201	VIc
22	SA38 ^f^	D	2017. 8. 1	Individual	+++	40.5	705	705	t529	VIb
23	SA45	D	2017. 11. 27	Individual	++	39.6	352	97	t1201	VIc
24	SA48	D	2018. 2. 4	Individual	+	38.8	705	705	t529	VIb
25	SA53	D	2018. 11. 18	Individual	++	39.0	352	97	t267	VIc
26	SA54	D	2018. 11. 20	Individual	+	39.2	705	705	t529	VIb
27	SA55	D	2018. 11. 27	Individual	+	40.5	352	97	t1201	VIc
28	SA56	D	2018. 12. 25	Individual	+	39.3	352	97	t267	VIc
29	SA61	D	2019. 8. 29	Individual	++	39.3	352	97	t1201	VIc
30	SA11 ^e^	E	2015. 9. 21	Individual	++	38.5	352	97	t267	VIc
31	SA18 ^e^	E	2016. 1. 18	Individual	+	39.3	352	97	t267	VIc
32	SA13	F	2015. 9. 27	Individual	+	38.5	NT1	705	t267	VIc
33	SA15	F	2015. 10. 13	Bulk	−	ND	NT2	97	t267	VIc
34	SA24 ^g^	F	2016. 4. 12	Individual	++	39.0	352	97	t267	VIc
35	SA25 ^g^	F	2016. 4. 12	Individual	++	39.0	352	97	t267	VIc
36	SA40	F	2017. 11. 7	Individual	−	ND	705	705	t529	VIb
37	SA41	F	2017. 11. 7	Individual	−	ND	352	97	t267	VIc
38	SA42	F	2017. 11. 7	Individual	++	ND	352	97	t267	VIc
39	SA43	F	2017. 11. 7	Individual	++	ND	352	97	t267	VIc
40	SA12	G	2015. 9. 21	Individual	++	ND	705	705	t529	VIb
41	SA26	G	2016. 7. 11	Individual	+++	40.0	352	97	t267	VIc
42	SA31 ^f^	G	2016. 11. 28	Individual	+++	41.5	352	97	t267	VIc
43	SA32	G	2017. 1. 10	Individual	+++	39.0	352	97	t359	VIc
44	SA33 ^f^	G	2017. 1. 19	Individual	+++	39.6	NT3	97	t267	VIc
45	SA34	G	2017. 2. 14	Individual	+	38.8	NT3	97	t359	VIc
46	SA46	G	2017. 12. 11	Individual	++	39.2	352	97	t267	VIc
47	SA47	G	2018. 1. 22	Individual	+	38.8	352	97	t865	VIc
48	SA49 ^f^	G	2018. 3. 1	Individual	++	40.5	352	97	t267	VIc
49	SA52	G	2018. 5. 27	Individual	±	39.3	705	705	t529	VIb
50	SA1	H	2015. 5. 20	Bulk	−	ND	352	97	t267	VIc
51	SA51	H	2018. 4. 23	Individual	+	39.5	352	97	t359	VIc
52	SA57	I	2019. 2. 23	Individual	+	39.2	705	705	t529	VIb
53	SA60	I	2019. 8. 30	Individual	++	39.0	705	705	t529	VIb
54	SA8	J	2015. 8. 13	Bulk	−	ND	705	705	t529	VIb
55	SA9	K	2015. 8. 13	Bulk	−	ND	NT3	97	t529	VIc
56	SA10	L	2015. 9. 5	Individual	++	40.5	352	97	t267	VIc
57	SA17	M	2015. 10. 18	Individual	+++	39.6	NT1	705	t529	VIb
58	SA37	N	2017. 6. 6	Individual	++	39.6	352	97	t359	VIc
59	SA50	O	2018. 3. 27	Individual	++	41.2	352	97	t359	VIc

^a^ Bulk, bulk tank milk; Individual, individual bovine sample with mastitis. ^b^ P.L tester was based on the modified California mastitis test. ^c^ ND, not determined. ^d^ isolated from the same cow in farm B on the same day. ^e^ recurrent mastitis bovine case. ^f^ peracute case. ^g^ isolated from the same cow in farm F on the same day.

**Table 2 microorganisms-10-02117-t002:** Correlation of genetic typing (CC and ST) and *spa* type.

CC	ST	*spa* Type					
		t267	t359	t529	t865	t1201	t3934
97	352	16 *^1^	4	0	1	3	0
	NT2	2 *^2^	1	1	0	0	0
398	398	0	0	0	0	0	6
	4894	0	0	0	0	0	1
705	705	0	0	20 *^3^	0	0	0
	NT1	1	0	1	0	2	0

*^1^ includes peracute mastitis case isolates SA31 and SA49. *^2^ includes peracute mastitis case isolate SA33. *^3^ includes peracute mastitis case isolate SA38.

**Table 3 microorganisms-10-02117-t003:** Profiles of PCR analysis for leukocidines, haemolysisns, and superantigenic toxins.

Group	*hla*, *hlb*, *hld*, *hlg*	*lukD*	*lukM*	*sec*	*seg*	*sei*	*sel2*	*sem*	*sen*	*seo*	*tst*	*lukS*, *lukF* *sea*, *seb* *sed*, *see* *seh*, *sej*	No. of Isolates
1	+	−	−	−	−	−	−	−	−	−	−	−	6
2	+	+	+	−	−	−	−	−	−	−	−	−	21 *^1^
3	+	+	+	−	−	−	−	−	−	+	−	−	10 *^2^
4	+	+	+	−	+	+	−	+	+	+	−	−	2
5	+	+	+	+	+	+	+	+	+	+	+	−	20 *^3^

*^1^ includes peracute mastitis case isolates SA31 and SA33. *^2^ includes peracute mastitis case isolate SA49. *^3^ includes peracute mastitis case isolate SA38.

**Table 4 microorganisms-10-02117-t004:** Prevalence data of genetic typing (CC, *coa*, and *spa*) and latex test.

CC	*coa* Type	*spa* Type	Latex Agglutination Test			
			−	+	+ +	+ + +
CC97	VIc	t267	0	3	12 *^1^	3 *^2^
		t359	0	0	2	3
		t529	0	1	0	0
		t865	0	0	1	0
		t1201	0	1	2	0
CC398	VII	t3934	0	0	1	6
CC705	VIb	t529	21 *^3^	0	0	0
		t1201	1	0	0	0
	VIc	t267	0	0	1	0
		t1201	0	1	0	0

*^1^ includes peracute mastitis case isolates SA33 and SA49. *^2^ includes peracute mastitis case isolate SA31. *^3^ includes peracute mastitis case isolate SA38.

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
