# Peer review of "Characterization of Staphylococcus aureus Isolates from Bovine Mastitis and Bulk Tank Milk: First Isolation of Methicillin-Susceptible Staphylococcus aureus in Japan"

_microorganisms, 2022, doi:10.3390/microorganisms10112117_

Round 1
Reviewer 1 Report
In this study, the authors described the distribution of S. aureus strains isolated from cows between 2015 to 2019 using sequence-based typing methods. This study is the first to isolate methicillin-sensitive ST398 strains in Japan, from a single farm out of 15 dairy farms surveyed in the central region of Miyagi Prefecture. The manuscript is well written, however, there are few comments that will need to be addressed before the final decision on the manuscript;
- The number of the collected samples is very low especially since the study duration is 4 years.
- Add table in the beginning of materials and methods to distribute your samples
- Please provide more details on the agar dilution method.
- be sure the name of the pathogens is italic throughout the manuscript
-There is no need for figure 1, please delete or move to supplementary materials.
- Remove the repeated results from the discussion section.
Author Response
Responses to the reviewer 1 (in italic):
In this study, the authors described the distribution of S. aureus strains isolated from cows between 2015 to 2019 using sequence-based typing methods. This study is the first to isolate methicillin-sensitive ST398 strains in Japan, from a single farm out of 15 dairy farms surveyed in the central region of Miyagi Prefecture. The manuscript is well written, however, there are few comments that will need to be addressed before the final decision on the manuscript;
Thank you for your positive and constructive comments that helped improve our manuscript. We have addressed each comment and the point-by-point responses have been given below (in italic, the line numbers are according to pdf file):
- The number of the collected samples is very low especially since the study duration is 4 years.
Thank you for your comment. Dr. Matsuda, one of the coauthors of this study, is a clinical veterinarian, not a researcher in a university. He collected S. aureus strains from dairy farms around his workplace, Livestock Medicine Training Center, Miyagi Prefecture Agricultural Mutual Aid Association, during his official work. This situation makes difficulty in collecting many isolates from bovines afflicted with mastitis in the study duration. Thus, the number of the isolates described in this study is relatively low.
- Add table in the beginning of materials and methods to distribute your samples
Thank you for your comment. We also think this comment is a good point. But, if we add the table of our samples (59 S. aureus isolates) in the beginning of the ‘Materials and Methods’ section, the table will have a few columns (isolate, farm, date of isolation, and source of milk). This information is almost the same as those of Table 1 shown in the ‘Results’ section of the original manuscript. We think this seems redundant. Thus, we do not present material’s table in the ‘Materials and Methods’ section.
- Please provide more details on the agar dilution method.
Thank you for your comment. We edited this part. Please see lines 185-191.
- be sure the name of the pathogens is italic throughout the manuscript.
Thank you for your comment. We checked the font of pathogens throughout the manuscript and edited according to the template of this journal, Microorganisms. Since the format of subsection, such as 3.1. Genetic Typing of S. aureus Isolates, is supposed to be in italic, thus the name of bacteria appeared in the subsection is written in roman type.
-There is no need for figure 1, please delete or move to supplementary materials.
Thank you for your comment. We agree with your comment. We therefore moved this figure to supplementary materials (Supplementary Figure 1).
- Remove the repeated results from the discussion section.
Thank you for your comments. A sentence in the Discussion section ‘During the survey period in this study (2015–2019), 53 strains were isolated from 9 farms, and 6 strains were isolated from 6 distinct farms in the middle region of Miyagi Prefecture.’ is a kind of redundant expression and is not important information in this study. We removed this description.
Reviewer 2 Report
This manuscript described methicillin-susceptible S. aureus isolates from bovine mastitis cases and bulk milk in Japan. Molecular characterization of the isolates was performed for S. aureus samples and ST398 strains were discussed. Although the data tables are well-presented, not all experiments were discussed or explained in discussion, also introduction part can be improved to give rationale for these experiments.
Please see the detailed comments below.
Although this manuscript emphasized the importance of the ST398 isolates, please change the title since from the 59 isolates, isolates belonged to ST398 are 7. Also, 49 isolates were from bovine mastitis case and 10 were from bulk milk. Please find a suitable title encompassing all these samples and experiments.
Line 6, 8: Please change one of them to keep using a consistent format. (Sendai 980-8572, Miyagi, 981-3602)
Line 9: equally -> equally.
Introduction: The rationale of the virulence factors (toxins) and tetracycline resistance gene test should be mentioned in the introduction. Was it for the classification or molecular characterization of the isolates? Please add a part explaining it.
Line 41-42: Although many people will agree on that statement considering the severity of the S. aureus infection, but please change the sentence (For example, is “the most” important one -> is “one of the most” important ones) or add more reasons for clarification to avoid the controversy. Considering the frequency of mastitis cases and recurrent infection, some people may think other bacteria such as E. coli or CNS is more important in bovine mastitis.
Line 63-64: Please change the sentence as the reference paper [9] said. (For example, …strains… belong to ST5 and ST121, respectively. -> ‘most’ strains ... belong to ST5 and ST121, respectively.)
Line 73: Suggestion: in this country -> so far.
Line 78 (m&m): Please add a part explaining how you selected MSSA samples. (For example, All S. aureus samples were MSSA? or you only included MSSA isolates in this study?)
Line 87-90: Please check the sentence again and modify it for better reading (For example, is -> showed).
Line 96-97: Please add a link or a reference information for the guidelines used for this study.
Line 98-110: Is there any reason for the authors used the method described in the reference [16]? It’s for anaerobic bacteria, and there are many other methods. Especially for S. aureus, lysostaphin is used in general. Please explain it in the response document.
Also, please describe the method more in detail such as volume of the sample or reagent used so the readers can repeat the method as described.
(Supplementary Table 1): Please add PCR product (amplicon) size and annealing temperature information in the table. For the multiplex PCR, primer set information should be included too.
Line 117-118: Please add the sequencing machine (or company) information for detailed description for the sequencing.
Table 1: Names in ‘Source of milk’ can be changed for better understanding (For example, bovine -> individual) Also, please explain the names with footnote (Bulk; bulk tank milk, individual; individual bovine sample with mastitis)
Date of isolation: The format for date can vary but can be confusing for some readers. Please indicate that information; year, month, day (For example, Date of isolation -> Date of isolation (YYYY.MM.DD.).
Line 158: Please check the name again (recurrence bovine case -> recurrent mastitis case).
Line 157, Line 160: Please check the ‘c’ and ‘f’ and modified them if they are different.
Line 169: Please add a reference and indicate the PUBMLST in the text as suggested by the website (Jolley et al. Wellcome Open Res 2018, 3:124 [version 1; referees: 2 approved].).
Line 198-200: Please describe ‘peracute’ more precisely. (For example, peracute isolates -> peracute mastitis case isolate(s))
Line 208: Significance or meaning of latex agglutination test should be explained in introduction or m&m part.
Line 225-228: Rationale or significance of the test should be explained in the previous part (introduction, m&m (To test XXX for XXX… AMR was tested).
Line 234-243: Antibiotic susceptibility is not the same as antimicrobial resistance gene detection and since this manuscript only tested tetracycline resistance genes, please make a separate paragraph (For example, ‘3.2.4. Tetracycline resistance genes detection’, ).
Line 240: Please modify the sentence for better reading. (For example, … genes in S. aureus strains isolated from mastitis cows via PCR analysis. -> … genes via PCR analysis in S. aureus strains isolated from mastitis cows.
Discussion: Latex agglutination test and detection of tetracycline resistance genes were not discussed. Please add a paragraph for each test explaining the results with comprehensive assessment with other research papers.
Also, isolates from recurrent infection case (SA31, SA33, SA38, SA49) should be discussed since they were highlighted throughout the manuscript. Please discuss their recurrent trait comparing with other isolates or other papers and find any possible reasons (many virulence factors such as MSCRAMMs).
Line 248: Please modify the sentence as line 150-151 to indicate that there were two types of sample origin (individual mastitis samples and bulk tank milk samples) or make the sentence brief since it is repeated again in conclusion.
Author Response
Responses to the reviewer 2 (in italic):
This manuscript described methicillin-susceptible S. aureus isolates from bovine mastitis cases and bulk milk in Japan. Molecular characterization of the isolates was performed for S. aureus samples and ST398 strains were discussed. Although the data tables are well-presented, not all experiments were discussed or explained in discussion, also introduction part can be improved to give rationale for these experiments.
Thank you for your insightful and constructive comments that helped improve our manuscript. According to your suggestion, we revised the original manuscript as follows. Please see our point-by-point responses given below (in italic, the line numbers are according to pdf file):
Please see the detailed comments below.
Although this manuscript emphasized the importance of the ST398 isolates, please change the title since from the 59 isolates, isolates belonged to ST398 are 7. Also, 49 isolates were from bovine mastitis case and 10 were from bulk milk. Please find a suitable title encompassing all these samples and experiments.
Thank you for your insightful suggestion. We agree with your comment. Thus, the title was changed to ‘Characterization of Staphylococcus aureus isolates from bovine mastitis and bulk tank milk: First isolation of methicillin-susceptible Staphylococcus aureus in Japan’.
Line 6, 8: Please change one of them to keep using a consistent format. (Sendai 980-8572, Miyagi, 981-3602)
Thank you for your comment. We edited as suggested (lines 7).
Line 9: equally -> equally.
Thank you for your comment. We edited as suggested (Lines 10).
Introduction: The rationale of the virulence factors (toxins) and tetracycline resistance gene test should be mentioned in the introduction. Was it for the classification or molecular characterization of the isolates? Please add a part explaining it.
Thank you for your thoughtful and constructive comment. We agree that this point is important. We therefor added description regarding the virulence factors in the Introduction section. Please see lines 73-79. The purpose of the experiment of tetracycline resistance gene was not for the classification of each isolates. It was to confirm the presence or absence of the gene(s) associated with tetracycline resistance phenotype. Thus, we did not mention about this point.
Line 41-42: Although many people will agree on that statement considering the severity of the S. aureus infection, but please change the sentence (For example, is “the most” important one -> is “one of the most” important ones) or add more reasons for clarification to avoid the controversy. Considering the frequency of mastitis cases and recurrent infection, some people may think other bacteria such as E. coli or CNS is more important in bovine mastitis.
Thank you for your important comment. We agreed with your comment. Thus, we edited as suggested (Lines 44).
Line 63-64: Please change the sentence as the reference paper [9] said. (For example, … strains… belong to ST5 and ST121, respectively. -> ‘most’ strains ... belong to ST5 and ST121, respectively.)
Thank you for your comment. It is important point. We edited as suggested (lines 71).
Line 73: Suggestion: in this country -> so far.
Thank you for your comment. We edited as suggested (lines 88).
Line 78 (m&m): Please add a part explaining how you selected MSSA samples. (For example, All S. aureus samples were MSSA? or you only included MSSA isolates in this study?)
Thank for the comment. Antibiotic susceptibility test for 59 isolates showed that all S. aureus samples were MSSA as shown in Supplementary Table 2. This was described in the Results section ‘All isolates were susceptible to oxacillin (MIC, 0.0625–0.5 µg/mL),--- (lines 407-410).
Line 87-90: Please check the sentence again and modify it for better reading (For example, is -> showed).
Thank you for your comment. We edited as suggested (lines 111).
Line 96-97: Please add a link or a reference information for the guidelines used for this study.
Thank you for your comment. As described in Materials and Methods, we have completely complied with the basic guidance and installation procedure rule. But unfortunately, these documents are only in Japanese. In addition, there is no link (URL information) of the description of Committee for Basic Guidelines for the Research Use of Industrial Animals in Miyagi Prefecture Agricultural Mutual Aid Association. We, therefore, cannot add information regarding this.
Line 98-110: Is there any reason for the authors used the method described in the reference [16]? It’s for anaerobic bacteria, and there are many other methods. Especially for S. aureus, lysostaphin is used in general. Please explain it in the response document.
Regarding your question, the reason of why we used the method (revised reference number 20) is just an economic issue in our laboratory. When we started this research project, our budget was not enough to prepare all reagents of high costs. Lysostaphin reagent in Japan is so expensive, thus, we searched other methods that use reagents with low costs other than lysostaphin. Since this method is not popular, we therefore described more precisely this protocol. Please see lines 126-139.
Also, please describe the method more in detail such as volume of the sample or reagent used so the readers can repeat the method as described.
Thank you for your comment. This comment is related to the previous one. We described the method more in detail. Please see our reply in the previous answer and lines 126-139.
(Supplementary Table 1): Please add PCR product (amplicon) size and annealing temperature information in the table. For the multiplex PCR, primer set information should be included too.
Thank you for your comment. We performed the experiments in an individual PCR reaction, not a multiplex PCR. Thus, we do not include the primer set information.
Line 117-118: Please add the sequencing machine (or company) information for detailed description for the sequencing.
Thank you for your comment. We added information of the sequencing machine (lines 147-148).
Table 1: Names in ‘Source of milk’ can be changed for better understanding (For example, bovine -> individual) Also, please explain the names with footnote (Bulk; bulk tank milk, individual; individual bovine sample with mastitis)
Thank you for your comment. We edited as suggested (Table 1 and its footnote line 311).
Date of isolation: The format for date can vary but can be confusing for some readers. Please indicate that information; year, month, day (For example, Date of isolation -> Date of isolation (YYYY.MM.DD.).
Thank you for your comment. We edited as suggested (Table 1).
Line 158: Please check the name again (recurrence bovine case -> recurrent mastitis case).
Thank you for your comment. We edited as suggested (Table 1 footnote line 315).
Line 157, Line 160: Please check the ‘c’ and ‘f’ and modified them if they are different.
Thank you for your comment. Isolates ‘c’ and ‘f’ are different, which were derived from different farms. We therefore edited accordingly (Table 1 footnote lines 314 and 317).
Line 169: Please add a reference and indicate the PUBMLST in the text as suggested by the website (Jolley et al. Wellcome Open Res 2018, 3:124 [version 1; referees: 2 approved].).
Thank you for your comment. We edited the website information in the text as your suggestion (line 324). In addition, we added a reference of this website (lines 650-651).
Line 198-200: Please describe ‘peracute’ more precisely. (For example, peracute isolates -> peracute mastitis case isolate(s))
Thank you for your comment. We edited as suggested (Table 2 footnote lines 353-355). In addition, we edited footnotes of Tables 3 and 4 (lines 373-375 and lines 410-403).
Line 208: Significance or meaning of latex agglutination test should be explained in introduction or m&m part.
Thank you for the important comment. We added description regarding latex agglutination test in the Introduction section. Please see lines 76-79 and 90-92.
Line 225-228: Rationale or significance of the test should be explained in the previous part (introduction, m&m (To test XXX for XXX… AMR was tested).
Thank you for your comment. This comment is related to your previous one ‘Introduction: The rationale of the virulence factors (toxins) and tetracycline resistance gene test---’. We edited this part as your suggestion (lines 55-58 and 90-91).
Line 234-243: Antibiotic susceptibility is not the same as antimicrobial resistance gene detection and since this manuscript only tested tetracycline resistance genes, please make a separate paragraph (For example, ‘3.2.4. Tetracycline resistance genes detection’, ).
Thank you for your comment. We agreed with your comment. But, we removed this Figure 1 and moved it to Supplementary Figure 1 as suggested by Reviewer 1. We therefore think that we do not need to make a separate subsection.
Line 240: Please modify the sentence for better reading. (For example, … genes in S. aureus strains isolated from mastitis cows via PCR analysis. -> … genes via PCR analysis in S. aureus strains isolated from mastitis cows.
Thank you for your comment. We edited as suggested (legend of Supplementary Figure 1). We moved this figure to supplementary data as suggested by Reviewer 1.
Discussion: Latex agglutination test and detection of tetracycline resistance genes were not discussed. Please add a paragraph for each test explaining the results with comprehensive assessment with other research papers.
Thank you for the important comment. We added a paragraph in the Discussion section describing latex agglutination test (lines 686-695). Regarding tetracycline resistance gene, as Reviewer 1 suggested to move Figure 1 to Supplementary materials, we deleted the Figure 1 in the main text. In addition, as I mentioned previously, the purpose of Figure 1 experiment is to confirm the presence/absence of the tetracycline gene, which is associated with the antibiotic resistance phenotype, thus, we did not mention regarding this.
Also, isolates from recurrent infection case (SA31, SA33, SA38, SA49) should be discussed since they were highlighted throughout the manuscript. Please discuss their recurrent trait comparing with other isolates or other papers and find any possible reasons (many virulence factors such as MSCRAMMs).
Thank you for the comment. You used the term ‘recurrent’ in this comment. But, we think this is ‘peracute’ based on the context of the comment in which four peracute cases are mentioned, SA31, SA33, SA38, and SA49. Thus, we added a paragraph in the Discussion section regarding the peracute cases (lines 686-695). For ‘recurrent cases’ we have described in the previous paragraph of Discussion (lines 658-685).
Line 248: Please modify the sentence as line 150-151 to indicate that there were two types of sample origin (individual mastitis samples and bulk tank milk samples) or make the sentence brief since it is repeated again in conclusion.
Thank you for your comment. We edited as you suggested (line 422).
Round 2
Reviewer 2 Report
The manuscript was revised well with responses to all comments and suggestions.
Please check again minor document editing needs like alignment of a paragraph (Line 191-197), Supplementary Figure 1 (Line 264) information (not included in supplementary materials (Line 381-384)), and typo (line 45: important one -> important ones).
There will be no more comments for the further revision except those.
Thank you.
Author Response
Responses to the reviewer 2 comments are as follows (in italic):
Please check again minor document editing needs like alignment of a paragraph (Line 191-197), Supplementary Figure 1 (Line 264) information (not included in supplementary materials (Line 381-384)), and typo (line 45: important one -> important ones).
Thank you for your positive comments and careful reviewing of our revised manuscript that helped improve our manuscript. We are sorry for our careless mistakes that you pointed out. Accordingly, we edited them as suggested. Please see the edited parts which are highlighted in yellow.